# Bandit Learning in Many-to-one Matching Markets with Uniqueness Conditions

## Abstract

An emerging line of research is dedicated to the problem of one-to-one matching markets with bandits, where the preference of one side is unknown and thus we need to match while learning the preference through multiple rounds of interaction. However, in many real-world applications such as online recruitment platform for short-term workers, one side of the market can select more than one participant from the other side, which motivates the study of the many-to-one matching problem. Moreover, the existence of a unique stable matching is crucial to the competitive equilibrium of the market. In this paper, we first introduce a more general new $\tilde{\alpha}$-condition to guarantee the uniqueness of stable matching in many-to-one matching problems, which generalizes some established uniqueness conditions such as *SPC* and *Serial Dictatorship*, and recovers the known $\alpha$-condition if the problem is reduced to one-to-one matching. Under this new condition, we design an MO-UCB-D4 algorithm with $O\left(\frac{NK\log(T)}{\Delta^2}\right)$ regret bound, where $T$ is the time horizon, $N$ is the number of agents, $K$ is the number of arms, and $\Delta$ is the minimum reward gap. Extensive experiments show that our algorithm achieves uniform good performances under different uniqueness conditions.

## 1 Introduction

The rise of platforms for the online matching market has led to an emergence of opportunities for companies to participate in personalized decision-making [14, 18]. Companies (like Thumbtack and Taskrabbit and Upwork platforms) use online platforms to address short-term needs or seasonal spikes in production demands, accommodate workers who are voluntarily looking for more flexible work arrangements or probation period before permanent employment. The supply and demand sides in two-sided markets make policies on the basis of their diversified needs, which is abstracted as a matching market with agent side and arm side, and each side has a preference profile over the opposite side. They choose from the other side according to preference and perform a matching. The stability of the matching result is a key property of the market [32, 1, 27].

The preferences in the online labor market may be unknown to one side in advance, thus matching while learning the preferences is necessary. The multi-armed bandit (MAB) [36, 13, 4] is an important tool for $N$ independent agents in matching market simultaneously selecting arms adaptively from received rewards at each round. The idea of applying MAB to one-to-one matching problems, introduced by [21], assumes that there is a central platform to make decisions for all agents. Following this, other works [22, 34, 7] consider a more general decentralized setting where there is no central platform to arrange matchings, and our work is also based on this setting.

However, it is not enough to just study the one-to-one setting. Take online short-term worker employment as an example, it is an online platform design with an iterative matching, where

employers have numerous similar short-term tasks or internships to be recruited. Workers can only choose one task according to the company's needs at a time while one company can accept more than one employee. Each company makes a fixed ranking for candidates according to its own requirements but workers have no knowledge of companies' preferences. The reward for workers is a comprehensive consideration of salary and job environment. Since tasks are short-term, each candidate can try many times in different companies to choose the most suitable job. We abstract companies as arms and workers as agents. Each arm has a *capacity q* which is the maximum number of agents this arm can accommodate. When an arm faces multiple choices, it accepts its most $q$ preferred agents. Agents thus compete for arms and may receive zero reward if losing the conflict. It is worth mentioning that arms with capacity $q$ in the many-to-one matching can not just be replaced by $q$ independent individuals with the same preference since there would be implicit competition among different replicates of this arm, not equal treatment. In addition, when multiple agents select one arm at a time, there may be no collision, which will hinder the communication among different agents under the decentralized assumption. They cannot distinguish who is more preferred by this arm in one round as it can accept more than one agent while this can be done in one-to-one case. Communication here lets each agent learn more about the preferences of arms and other agents, so as to formulate better policies to reduce collisions and learn fast about their stable results.

This work focuses on a many-to-one market under uniqueness conditions. Previous work [10, 15] emphasize the importance of constructing a unique stable matching for the equilibrium of matching problems and some existing uniqueness conditions are studied in many-to-one matching, such as *Sequential Preference Condition (SPC)* and *Acyclicity* [26, 2]. Our work is motivated by [7], but the unique one-to-one mapping between arms and agents in their study which gives a surrogate threshold for arm elimination does not work in the many-to-one setting. And the uniqueness conditions in many-to-one matching are not well-studied, which also brings a challenge to identify and leverage the relationship between the resulting stable matching and preferences of two sides in the design of bandit algorithms. We propose an $\tilde{\alpha}$-condition that can guarantee a unique stable matching and recover $\alpha$-condition [19] if reduced to the one-to-one setting. We establish the relationships between our new $\tilde{\alpha}$-condition and existing uniqueness conditions in many-to-one setting.

In this paper, we study the bandit algorithm for a decentralized many-to-one matching market with uniqueness conditions. Under our newly introduced $\tilde{\alpha}$-condition, we design an MO-UCB-D4 algorithm with arm elimination and the regret can be upper bounded by $O\left(\frac{NK \log(T)}{\Delta^2}\right)$, where $N$ is the number of agents, $K$ is the number of arms, and $\Delta$ is the minimum reward gap. Finally, we conduct a series of experiments to simulate our algorithm under various conditions of *Serial dictatorship*, *SPC* and $\tilde{\alpha}$-*condition* to study the stability and regret of the algorithm.

**Related Work** The study of matching markets has a long history in economics and operation research [8, 6, 32] with real applications like school enrollment, labor employment, hospital resource allocation, and so on [1, 23, 31, 17]. A salient feature of market matching is making decisions for competing players on both sides [36, 12]. MAB is an important tool to study matching problems under uncertainty to obtain a maximum reward, and upper confidence bound algorithm (UCB) [4] is a typical algorithm, which sets a confidence interval to represent uncertainty. Matching market with MAB is studied in both centralized and decentralized setting [21, 22]. Following these, Abishek Sankararaman et al. [34] propose a phased UCB algorithm under a uniqueness condition, *Serial Dictatorship*, to manage collisions. They solve the problem of the decentralized market without knowing arm-gaps or time horizon, and reduce the probability of linear regret through non-monotonic arm elimination. The introduction of the uniqueness condition plays an important role in the equilibrium of matching results [15, 7]. Under a stronger and robust condition, Uniqueness Consistency [19], Soumya Basu et.al [7] apply MAB to online matching and obtain robust results that the subset of stable matchings being separated from the system does not affect other stable matchings.

We discuss many-to-one problems such as online short-term employment and MOOC [14, 24, 18] as the one-to-one setting has limitations in practice. Somouaoga Bonkoungo [9] runs a student-proposing deferred acceptance algorithm (DA) [12] to study decentralized college admission. Ahmet Altinok [3] considers dynamic matching in many-to-one that can be solved as if it is static many-to-one or dynamic one-to-one under certain assumptions. As the existence and uniqueness of competitive equilibrium and core are important to allocations, the unique stable results need to be considered [27]. Similar to conditions for unique stable matching in one-to-one, some uniqueness conditions of stable results in the many-to-one setting also are studied [16, 28, 15, 2, 27].

## 2 Setting

This paper considers a many-to-one matching market $\mathcal{M} = (\mathcal{K}, \mathcal{J}, \mathcal{P})$, where $\mathcal{K} = [K], \mathcal{J} = [N]$ are a finite arm set and a finite agent set, respectively. And each arm $k$ has a capacity $q_k \geq 1$. To guarantee that no agents will be unmatched, we focus on the market with $N \leq \sum_{i=1}^{K} q_i$. $\mathcal{P}$ is the fixed preference order of agents and arms, which is ranked by the mean reward. We assume that arm preferences for agents are unknown and needed to be learned. If agent $j$ prefers arm $k$ over $k'$, which also means that $\mu_{j,k} > \mu_{j,k'}$, we denote by $k \succ_j k'$. And the preference is strict that $\mu_{j,k} \neq \mu_{j,k'}$ if $k \neq k'$. Similarly, each arm $k$ has a fixed and known preference $\succ_k$ over all agents, and specially, $j \succ_k j'$ means that arm $k$ prefers agent $j$ over $j'$. Throughout, we focus on the market where all agent-arm pairs are *mutually acceptable*, that is, $j \succ_k \emptyset$ and $k \succ_j \emptyset$ for all $k \in [K]$ and $j \in [N]$.

Let mapping $m$ be the matching result. $m_t(j)$ is the matched arm for agent $j$ at time $t$, and $\gamma_t(k)$ is the agents set matched with arm $k$[1]. Every time agent $j$ selects an arm $I_t(j)$, and we use $M_t(j)$ to denote whether $j$ is successfully matched with its selected arm. $M_t(j) = 1$ if agent $j$ is matched with $I_t(j)$, and $M_t(j) = 0$, otherwise. If multiple agents select arm $k$ at the same time, only top $q_k$ agents can successfully match. The agent $j$ matched with arm $k$ can observe the reward $X_{j,m_t(j)}(t)$, where the random reward $X_{j,k}(t) \in [0, 1]$ is independently drawn from a fixed distribution with mean $\mu_{j,k}$. While the unmatched ones have collisions and receive zero reward. Generally, the reward obtained by agent $j$ is $X_{j,I_t(j)}(t) M_t(j)$.

An agent $j$ and an arm $k$ form a *blocking pair* for a matching $m$ if they are not matched but prefer each other over their assignments, i.e. $k \succ_j m(j)$ and $\exists j' \in \gamma(k), j \succ_k j'$. We say a matching satisfies individually rationality (IR), if $a_j \succ_{p_i} \emptyset$ and $p_i \succ_{a_j} \emptyset$ for all $i \in [N]$ and $j \in [K]$, that is, every worker prefers to find a job rather than do nothing, and every company also wants to recruit workers rather than not recruit anyone. Under the IR condition, a matching in the many-to-one setting is *stable* if there does not exist a blocking pair [33, 35].

This paper considers the matching markets under the uniqueness condition. Thus the overall goal is to find the unique stable matching between the agent side and arm side through iterations. Let $m^*(j)$ be the stable matched arm for agent $j$ under the stable matching $m^*$. The reward obtained by agent $j$ is compared against the reward received by matching with $m^*(j)$ at each time. We aim to minimize the expected stable regret for agent $j$ over time horizon $T$, which is defined as

$$R_j(T) = T\mu_{j,m^*(j)} - \mathbb{E}\left[\sum_{t=1}^{T} M_t(j)X_{j,I_t(j)}(t)\right].$$

## 3 Algorithm

In this section, we introduce our MO-UCB-D4 Algorithm (Many-to-one UCB with Decentralized Dominated arms Deletion and Local Deletion Algorithm) (Algorithm 1) for the decentralized many-to-one market, where there is no platform to arrange actions for agents, which leads to conflicts among agents. The MO-UCB-D4 algorithm for each agent $j$ first takes agent set $\mathcal{J}$ and arm set $\mathcal{K}$ as input and chooses a parameter $\theta \in (0, 1/K)$ (discussed in Section C). It sets multiple phases, and each phase $i$ mainly includes regret minimization block (line 6 - 12) and communication block (line 13 - 16) with duration $2^{i-1}, i = 1, 2, \cdots$.

For each agent $j$ in phase $i$, the algorithm adds arm deletion to reduce potential conflicts, which mainly contains global deletion and local deletion. The former eliminates the arms most preferred by agents who rank higher than agent $j$ and obtain active set $\mathrm{Ch}_j[i]$ (line 4), and the latter deletes the arms that still have many conflicts with agent $j$ after global deletion (line 6). We set a collision counter $C_{j,k}[i]$ to record the number of collisions for agent $j$ pulling arm $k$.

In regret minimization block of phase $i$, we use $L_j[i] = \{k : C_{j,k}[i] \geq \lceil \theta 2^i \rceil\}$ to represent the arms that collide more times than a threshold $\lceil \theta 2^i \rceil$ when matching with agent $j$. Arms in $L_j[i]$ are first locally deleted to reduce potential collisions for agent $j$ (line 6). After that, agent $j$ selects an optimal action $I_t(j)$ from remaining arms in $\mathrm{Ch}_j[i] \setminus L_j[i]$ in phase $i$ according to UCB index, which is computed by $\hat{\mu}_{j,k}(t-1) + \sqrt{\frac{2\alpha \log(t)}{N_{j,k}(t-1)}}$ (line 7), where $N_{j,k}(t-1)$ is the number that agent $j$ and arm

---
[1]The mapping $m$ is not reversible as it is not a injective, thus we do not use $m_t^{-1}(k)$.

**Algorithm 1** MO-UCB-D4 algorithm (for agent $j$)

**Input:**

    $\theta \in (0, 1/K), \alpha > 1$.

1: Set global dominated set $G_j[0] = \phi$

2: **for** phase $i = 1, 2, ...$ **do**

3:     Reset the collision set $C_{j,k}[i] = 0, \forall k \in [K]$;

4:     Reset active arms set $\texttt{Ch}_j[i] = [K] \backslash G_j[i-1]$;

5:     **if** $t < 2^i + NK(i-1)$ **then**

6:         Local deletion $L_j[i] = \{k : C_{jk}[i] \geq \lceil \theta 2^i \rceil\}$;

7:         Play arm $I_t(j) \in \underset{k \in \texttt{Ch}_j[i] \backslash L_j[i]}{\arg \max} \left( \hat{\mu}_{j,k}(t-1) + \sqrt{\frac{2\alpha \log(t)}{N_{j,k}(t-1)}} \right)$;

8:         **if** $k = I_t(j)$ is successfully matched with agent $j$, i.e. $m_t(j) = k$ **then**

9:             Update estimate $\hat{\mu}_{j,k}(t)$ and matching count $N_{j,k}(t)$;

10:        **else**

11:           $C_{j,k}[i] = C_{j,k}[i] + 1$;

12:        **end if**

13:     **else if** $t = 2^i + NK(i-1)$ **then**

14:        $\mathcal{O}_j[i] \leftarrow$ most matched arm in phase $i$;

15:        $G_j[i] \leftarrow COMMUNICATION(i, \mathcal{O}_j[i])$;

16:     **end if**

17: **end for**

$k$ have been matched at time $t - 1$. If the selected arm is successfully matched with agent $j$, then the algorithm updates estimated reward $\hat{\mu}_{j,k}(t) = \frac{1}{N_{j,k}(t)} \sum_{s=1}^{t} 1\{I_s(j) = k \text{ and } M_s(j) = 1\} X_{j,k}(t)$ and $N_{j,k}(t)$ (line 9). Otherwise, the collision happens (line 11) and $j$ receives zero reward. The regret minimization block identifies the most played arm $\mathcal{O}_j[i]$ for agent $j$ in each phase $i$, which is estimated as the best arm for $j$, thus making optimal policy to minimize expected regret.

**Algorithm 2** COMMUNICATION

**Input:**

    Phase number $i$, and most played arms $\mathcal{O}_j[i]$ for agent $j$, $\forall j \in [N]$.

1: Set $\mathcal{C} = \emptyset$;

2: **for** $t = 1, 2, \cdots, NK - 1$ **do**

3:     **if** $K(j-1) \leq t \leq Kj - 1$ **then**

4:         Agent $j$ plays arm $I_t(j) = (t \mod K) + 1$;

5:         **if** Collision Occurs **then**

6:             $\mathcal{C} = \mathcal{C} \cup \{I_t(j)\}$;

7:         **end if**

8:     **else**

9:         Play arm $I_t(j) = \mathcal{O}_j[i]$;

10:     **end if**

11: **end for**

12: RETURN $\mathcal{C}$;

In the communication block (Algorithm 2), there are $N$ sub-blocks, each with duration $K$. In the $\ell - th$ sub-block, only agent $\ell$ pulls arm 1, arm 2, $\cdots$, arm $K$ in round-robin while the other agents select their most preferred arms estimated as the most played ones (line 4). This block aims to detect globally dominated arms for agent $j$: $G_j[i] \subset \{\mathcal{O}_{j'}[i] : j' \succ_{\mathcal{O}_{j'}[i]} j\}$. Under stable matching $m^*$, the globally dominated arms set for agent $j$ is denoted as $G_j^*$. After the communication block in phase $i$, each agent $j$ updates its active arms set $\texttt{Ch}_j[i+1]$ for phase $i+1$, by globally deleting arms set $G_j[i]$, and enters into the next phase (line 4 in Algorithm 1).

Hence, multi-phases setting can guarantee that the active set in different phases has no inclusion relationship so that if an agent deletes an arm in a certain phase, this arm can still be selected in the later rounds. This ensures that each agent will not permanently eliminate its stable matched arm, and when the agent mistakenly deletes an arm, it will not lead to linear regret.

## 4 Results

### 4.1 Uniqueness Conditions

#### 4.1.1 $\tilde{\alpha}$-condition

Constructing a unique stable matching plays an important role in market equilibrium and fairness [10, 15]. With uniqueness, there would be no dispute about adopting stable matching preferred by which side, thus it is more fair. When the preferences of agents and arms are given by some utility functions instead of random preferences, like the payments for workers in the labor markets, the stable matching is usually unique. Thus the assumption of the unique stable matching is quite common in real applications. In this section, we propose a new uniqueness condition, $\tilde{\alpha}$-condition. First, we introduce *uniqueness consistency (Unqc)* [19], which guarantees robustness and uniqueness of markets.

**Definition 1.** *A preference profile satisfies uniqueness consistency if and only if*

*(i) there exists a unique stable matching $m^*$;*

*(ii) for any subset of arms or agents, the restriction of the preference profile on this subset with their stable-matched pair has a unique stable matching.*

It guarantees that even if an arbitrary subset of agents are deleted out of the system with their respective stable matched arms, there still exists a unique stable matching among the remaining agents and arms. This condition allows any algorithm to identify at least one stable pair in a unique stable matching system and guides the system to a global unique stable matching in an iterative manner. To obtain consistent stable results in the many-to-one market, we propose a new $\tilde{\alpha}$-*condition*, which is a sufficient and necessary condition for Unqc (proved in Appendix B).

We considers a finite set of arms $[K] = \{1, 2, \cdots, K\}$ and a finite set of agents $[N] = \{1, 2, \cdots, N\}$ with preference profile $\mathcal{P}$. Assume that $[N]_r = \{A_1, A_2, \cdots, A_N\}$ is a permutation of $\{1, 2, \cdots, N\}$ and $[K]_r = \{c_1, c_2, \cdots, c_K\}$ is a permutation of $\{1, 2, \cdots, K\}$. Denote $[N], [K]$ as the left order and $[N]_r, [K]_r$ as the right order. The $k$-th arm in the right order set $[K]_r$ has the index $c_k$ in the left order set $[K]$ and the $j$-th agent in the right order set $[N]_r$ has the index $A_j$ in the left order set $[N]$. Considering arm capacity, we denote $\gamma^*(c_k)$ (right order) as the stable matched agents set for arm $c_k$.

**Definition 2.** *A many-to-one matching market satisfies the $\tilde{\alpha}$-condition if,*

*(i) The left order of agents and arms satisfies*

$$\forall j \in [N], \forall k > j, k \in [K], \mu_{j,m^*(j)} > \mu_{j,k},$$

*where $m^*(j)$ is agent $j$'s stable matched arm;*

*(ii) The right order of agents and arms satisfies*

$$\forall k < k' \leq K, c_k \in [K]_r, A_{k'} \subset [N]_r, \gamma^*(c_k) \succ_{c_k} A_{\sum_{i=1}^{k'-1} q_{c_i}+1},$$

*where the set $\gamma^*(c_k)$ is more preferred than $A_{\sum_{i=1}^{k'-1} q_{c_i}+1}$ means that the least preferred agent in $\gamma^*(c_k)$ for $c_k$ is better than $A_{\sum_{i=1}^{k'-1} q_{c_i}+1}$ for $c_k$.*

Under our $\tilde{\alpha}$-*condition*, the left order and the right order satisfy the following rule. The left order gives rankings according to agents' preferences. The first agent in the left order set $[N]$ prefers arm 1 in $[K]$ most and has it as the stable matched arm. Similar properties for the agent 2 to $q_1$ since arm 1 has $q_1$ capacity. Then the $(q_1 + 1)$-th agent in the left order set $[N]$ has arm 2 in $[K]$ as her stable matched arm and prefers arm 2 most except arm 1. The remaining agents follow similarly. Similarly, the right order gives rankings according to arms' preferences. The first arm 1 in the right order set $[K]_r$ most prefers first $q_{c_1}$ agents in the right order set $[N]_r$ and takes them as its stable matched agents. The remaining arms follow similarly.

This condition is more general than existing uniqueness conditions like *SPC* [28] and can recover the known $\alpha$-condition in one-to-one matching market [19]. The relationship between the existing uniqueness conditions and our proposed conditions will be analyzed in detail later in Section 4.1.2.

The main idea from one-to-one to many-to-one analysis is to replace individuals with sets. In general, under $\tilde{\alpha}$-*condition*, the left order satisfies that when arm 1 to arm $k - 1$ are removed, agents

$\left(\sum_{i=1}^{k-1} q_i + 1\right)$ to $\left(\sum_{i=1}^{k} q_i\right)$ prefer $k$ most, and the right order means that when $A_1$ to agents $A_{\sum_{i=1}^{k-1} q_i}$ are removed, arm $k$ prefers agents $\mathcal{A}_k = \{A_{\sum_{i=1}^{k-1} q_{c_i}+1}, A_{\sum_{i=1}^{k-1} q_{c_i}+2}, \cdots, A_{\sum_{i=1}^{k} q_{c_i}}\}$, where $\mathcal{A}_k$ is the agent set that are most $q_k$ preferred by arm $k$ among those who have not been matched by arm $1, 2, \cdots, k-1$. Te next theorem give a summary.

**Theorem 1.** *If a market* $\mathcal{M} = (\mathcal{K}, \mathcal{J}, \mathcal{P})$ *satisfies* $\tilde{\alpha}$*-condition, then* $m^*(\sum_{i=1}^{j-1} q_i + 1) = m^*(\sum_{i=1}^{j-1} q_i + 2) = \cdots = m^*(\sum_{i=1}^{j} q_i) = j$ *(the left order),* $\gamma^*(c_k) = \mathcal{A}_k$ *and* $m^*(\mathcal{A}_j) = c_j$ *(the right order) under stable matching.*

Under $\tilde{\alpha}$-condition, the stable matched arm may not be the most preferred one for each agent $j$, $j \in [N]$, thus (i) we do not have $m^*(j)$ to be dominated only by the agent 1 to agent $j-1$, i.e. there may exist $j' > j$, s.t. $j' \succ_{m^*(j)} j$; (ii) the left order may not be identical to the right order, we define a mapping *lr* to match the index of an agent in the left order with the index in the right order, i.e. $A_{lr(j)} = j$. From Theorem 1, the stable matched set for arm $k$ is its first $q_k$ preferred agents $\gamma^*(c_k) = \mathcal{A}_k$. We define $lr$ as $lr(i) = \max\{j : A_j \in \gamma^*(m^*(i)), j \in [N]\}$, that is, in the right order, the mapping for arm $k \in [K]$ is the least preferred one among its most $q_k$ preferred agents. Note that this mapping is not an injective, i.e. $\exists j, j'$, s.t. agent $j = A_{lr(j)} = A_{lr(j')}$. An intuitive representation can be seen in Figure 4 in Appendix A.1.

### 4.1.2 Unique Stable Conditions in Many-to-one Matching

Uniqueness consistency (Unqc) leads the stable matching to a robust one which is a desirable property in large dynamic markets with constant individual departure [7]. A precondition of Unqc is to ensure global unique stability, hence finding uniqueness conditions is essential.

The existing unique stable conditions are well established in one-to-one setting (analysis can be found in Appendix B), and in this section, we focus on uniqueness conditions in many-to-one market, such as *SPC*, [28], *Aligned Preference*, *Serial Dictatorship Top-top match* and *Acyclicity* [26, 2, 28] (Definition 9, 7, 8, 10 in Appendix B.2). Takashi Akahoshi [2] proposes a necessary and sufficient condition for uniqueness of stable matching in many-to-one matching where unacceptable agents and arms may exist on both sides. We denote their condition as *Acyclicity*$^*$. Under our setting, both two sides are acceptable, and we first give the proof of that *Acyclicity*$^*$ is a necessary and sufficient condition for uniqueness in this setting (see Section B.2.4 in Appendix B). We then give relationships between our newly $\tilde{\alpha}$-condition and other existing uniqueness conditions, intuitively expressed in Figure 1, and we give proof for this section in Appendix B.2.

**Lemma 1.** *In a many-to-one matching market* $\mathcal{M} = (\mathcal{K}, \mathcal{J}, \mathcal{P})$*, both Serial Dictatorship and Aligned Preference can produce a unique stable matching and they are equivalent.*

**Theorem 2.** *In a many-to-one matching market* $\mathcal{M} = (\mathcal{K}, \mathcal{J}, \mathcal{P})$*, our* $\tilde{\alpha}$*-condition satisfies:*

*(i) SPC is a sufficient condition to* $\tilde{\alpha}$*-condition;*

*(ii)* $\tilde{\alpha}$*-condition is a necessary and sufficient condition to Unqc;*

*(iii)* $\tilde{\alpha}$*-condition is a sufficient condition to* $Acyclicity^*$*.*

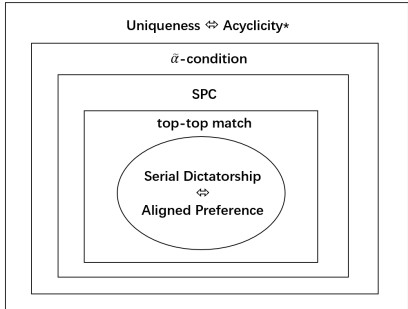

Figure 1: Relations of Uniqueness Conditions in Many-to-one Market.

## 4.2 Theoretical Results of Regret

We then provide theoretical results of MO-UCB-D4 algorithm under our $\tilde{\alpha}$-condition. Recall that $G_j^*$ is the globally dominated arms for agent $j$ under stable matching $m^*$. For each arm $k \notin G_j^*$, we give the definition of the *blocking agents* for arm $k$ and agent $j$: $\mathcal{B}_{jk} = \{j' : j' \succ_k j, k \notin G_j^*\}$, which contains agents more preferred by arm $k$ than $j$. The *hidden arms* for agent $j$ is $\mathcal{H}_j = \{k : k \notin G_j^*\} \cap \{k : \mathcal{B}_{jk} \neq \emptyset\}$. The reward gap for agent $j$ and arm $k$ is defined as $\Delta_{jk} = |\mu_{j,m^*(j)} - \mu_{j,k}|$ and the minimum reward gap across all arms and agents is $\Delta = \min_{j \in [N]}\{\min_{k \in [K]} \Delta_{j,k}\}$. We assume that the reward is different for each agent, thus $\Delta_{j,k} > 0$ for every agent $j$ and arm $k$.

**Theorem 3.** *(Regret upper bound) Let $J_{\max}(j) = \max\{j + 1, \{j' : \exists k \in \mathcal{H}_j, j' \in \mathcal{B}_{jk}\}\}$ be the max blocking agent for agent $j$ and $f_{\tilde{\alpha}}(j) = j + lr_{\max}(j)$ is a fixed factor depends on both the left order and the right order for agent $j$. Following MO-UCB-D4 algorithm with horizon $T$, the expected regret of a stable matching under $\tilde{\alpha}$-condition (Definition 2) for agent $j \in [N]$ is upper bounded by*

$$
\mathbb{E}\left[R_j(T)\right] \leq \sum_{k \notin G_j^* \cup m^*(j)} \frac{8\alpha}{\Delta_{jk}}\left(\log(T) + \sqrt{\frac{\pi}{\alpha}\log(T)}\right) + \sum_{k \notin G_j^*}\sum_{j' \in \mathcal{B}_{jk}: k \notin G_{j'}^*} \frac{8\alpha\mu_{j,m^*(j)}}{\Delta_{j'k}^2}\left(\log(T) + \sqrt{\frac{\pi}{\alpha}\log(T)}\right)
$$

$$
+ c_j \log_2(T) + O\left(\frac{N^2 K^2}{\Delta^2} + \left(\min(1, \theta|\mathcal{H}_j|)f_\alpha(J_{\max}(j)) + f_{\tilde{\alpha}}(j) - 1\right)2^{i^*} + N^2 K i^*\right),
$$

*where $i^* = \max\{8, i_1, i_2\}$ (then $i^* \leq 8$ and $i_1, i_2$ are defined in equation (3)), and $lr_{\max}(j) = \max\{lr(j') : 1 \leq j' \leq j\}$, is the maximum right order mapping for agent $j'$ who ranks higher than $j$.*

From Theorem 3, the scale of the regret upper bound under $\tilde{\alpha}$-*condition* is $O\left(\frac{NK\log(T)}{\Delta^2}\right)$ and the proof is in Section 3.

**Proof Sketch of Theorem 3.** Under $\tilde{\alpha}$-*condition*, we only need to discuss the regret of the unique result. We construct a *good phase* (in Appendix A.2) and denote that the time point of agent $j$ reaching its *good phase* by $\tau_j$. After $\tau_j$, agent $j$ could identify its best arm and matches with his stable pair. Thus, from phase $\tau_j$ on-wards, agent $j + 1$ will find the set of globally dominated arms $G_{j+1}^*$ and will eliminate arm $m^*(j)$ if $m^*(j)$ brings collisions in communication block according to Algorithm 1. Global deletion here follows the left order. Then when agent $j$ enters into regret minimization block next phase, the times it plays a sub-optimal arm is small which leads to a small total number of collisions experienced by agent $j + 1$. Then the process of each agent after *good phase* is divided into two stages: before $\tau_j$ and after $\tau_j$. After $\tau_j$, according to the causes of regret, it is divided into four blocks: collision, local deletion, communication, and sub-optimal play. Phases before $\tau_j$ can be bounded by induction. The regret decomposition is bound by the following.

**Lemma 2.** *(Regret Decomposition) For a stable matching under $\tilde{\alpha}$-condition, the upper bound of regret for the agent $j \in [N]$ under our algorithm can be decomposed by:*

$$
\mathbb{E}\left[R_j(T)\right] \leq \underbrace{\mathbb{E}\left[S_{F_{\alpha j}}\right]}_{\text{(Regret before phase } F_{\alpha j}\text{)}} + \underbrace{\min(\theta|\mathcal{H}_j|, 1)\mathbb{E}\left[S_{V_{\alpha j}}\right]}_{\text{(Local deletion)}} + \underbrace{\left((K - 1 + |\mathcal{B}_{j,m^*(j)}|)\log_2(T) + NK\mathbb{E}\left[V_{\alpha j}\right]\right)}_{\text{(Communication)}}
$$

$$
+ \underbrace{\sum_{k \notin G_j^*}\sum_{j' \in \mathcal{B}_{j,k}: k \notin G_{j'}^*} \frac{8\alpha\mu_{j,m^*(j)}}{\Delta_{j',k}^2}\left(\log(T) + \sqrt{\frac{\pi}{\alpha}\log(T)}\right)}_{\text{(Collision)}}
$$

$$
+ \underbrace{\sum_{k \notin G_j^* \cup m^*(j)} \frac{8\alpha}{\Delta_{j,k}}(\log(T) + \sqrt{\frac{\pi}{\alpha}\log(T)})}_{\text{(Sub-optimal play)}} + NK\left(1 + (\phi(\alpha) + 1)\frac{8\alpha}{\Delta^2}\right),
$$

*where $F_{\alpha j}, V_{\alpha j}$ are the time points when agent $j$ enters into $\tilde{\alpha}$-Good phase and $\tilde{\alpha}$-Low Collision phase respectively, mentioned as "good phase" above, are defined in Appendix A.2.*

## 5   Difficulties and Solutions

While putting forward our $\tilde{\alpha}$-condition in the many-to-one setting, many new problems need to be taken into account.

**From one-to-one setting to many-to-one setting**   First, although we assume that arm preference is over individuals rather than combination of agents, the agents matched by one arm are not independent. Specially, arms with capacity $q$ can not just be replaced by $q$ independent individuals with the same preference. Since there would be implicit competition among different replicates of this arm, and it can reject the previously accepted agents when it faces a more preferred agent. Secondly, collisions among agents is one of main causes of regret in decentralized setting, while capacity will hinder the collision-reducing process. In communication block, when two agents select one arm at a time, as an arm can accept more than one agent, these two cannot distinguish who is more preferred by this arm, while it can be done in one-to-one markets. Thus it is more difficult to identify arm preferences for each agent. The $lr$ in [7] is a one-to-one mapping that corresponds the agent index in the left order and the agent index in the right order, which is related to regret bound (Theorem 3 in [7] and Theorem 3 in our work). While it does not hold in our setting. To give a descriptive range of matched result for each arm under $\tilde{\alpha}$-condition, we need to define a new mapping.

In order to solve these problems, we explain as follows: First, since capacity influence the communication among agents, we add communication block and introduce an arm set $G_j^*$, which will be deleted before each phase to reduce collisions, where $G_j^*$ contains arms that will block agent $j$ globally under stable matching $m^*$. Second, the idea from one-to-one to many-to-one is a transition from individual to set. It is natural to split sets into individuals or design a bridge to correspond sets to individuals. We construct a new mapping $lr$ (Figure 4 in Appendix A) from agent $j$ in the left order to agents in the right order under $\tilde{\alpha}$-condition. $lr$ maps each arm $k$ to the least preferred one of its stable matched agents in the right order, thus giving a matching between individuals and individuals and constructing the range of the stable matched agents set (Theorem 1). Except $lr$, capacity also influences regret mainly in communication block, as mentioned in the first paragraph.

**From $\alpha$-condition to $\tilde{\alpha}$-condition**   To extend $\alpha$-condition to the many-to-one setting, it needs to define preferences among sets. However, there might be exponential number of sets due to the combinatorial structure and simply constraining preferences over all possible sets will lead to high complexity. Motivated by $\alpha$-condition which characterizes properties of matched pairs in one-to-one setting, we come up with a possible constraint by regarding the arm and its least preferred agent in the matched set as the *matched pair* and define preferences according to this grouping. It turns out that we only need to define the preferences of arms over disjoint sets of agents to complete the extension as $\alpha$-condition is defined under the stable matching, which can also fit the regret analysis well. As a summary, there might be other possible ways to extend the $\alpha$-condition but we present a successful trial to not only give a good extension with similar inclusion relationships but also guarantee good regret bound.

## 6   Experiments

In this section, we verify the experimental results of our MO-UCB-D4 algorithm (Algorithm 1) for decentralized many-to-one matching markets. For all experiments, the rankings of all agents and arms are sampled uniformly. We set the reward value towards the least preferred arm to be $1/N$ and the most preferred one as 1 for each agent, then the reward gap between any adjacently ranked arms is $\Delta = 1/N$. The reward for agent $j$ matches with arm $k$ at time $t$ $X_{j,k}(t)$ is sampled from $\mathrm{Ber}(\mu_{j,k})$. The capacity is equally set as $q = N/K$. We investigate how the cumulative regret and cumulative market unstability depend on the size of the market and the number of arms under three different unique stability conditions: *Serial Dictatorship*, *SPC*, $\tilde{\alpha}$-*condition*. The former cumulative regret is the total mean reward gap between the stable matching result and the simulated result, and the latter cumulative unstability is defined as the number of unstable matchings in round $t$. In our experiments, all results are averaged over 10 independent runs, hence the error bars are calculated as standard deviations divided by $\sqrt{10}$.

**Varying the market size**   To test effects on two indicators, cumulative regret and cumulative unstability, we first varying $N$ with fixed $K$ with market size of $N \in \{10, 20, 30, 40\}$ agents

319 and $K = 5$ arms. The number of rounds is set to be $100,000$. The cumulative regret in Figure
320 2(a)(c)(e) show an increasing trend with convergence as the number of agents increases under these
321 three conditions. When the number of agents increases, there is a high probability of collisions
322 among different agents, resulting in the increase of cumulative regret. Similar results for cumulative
323 unstability are shown in Figure 2(b)(d)(f). When $N$ is larger, the number of unstable pairs becomes
324 more. With the increase of the number of rounds, both two indicators increase first and then tend to
325 be stable. The jumping points are caused by multi-phases setting of MO-UCB-D4 algorithm.

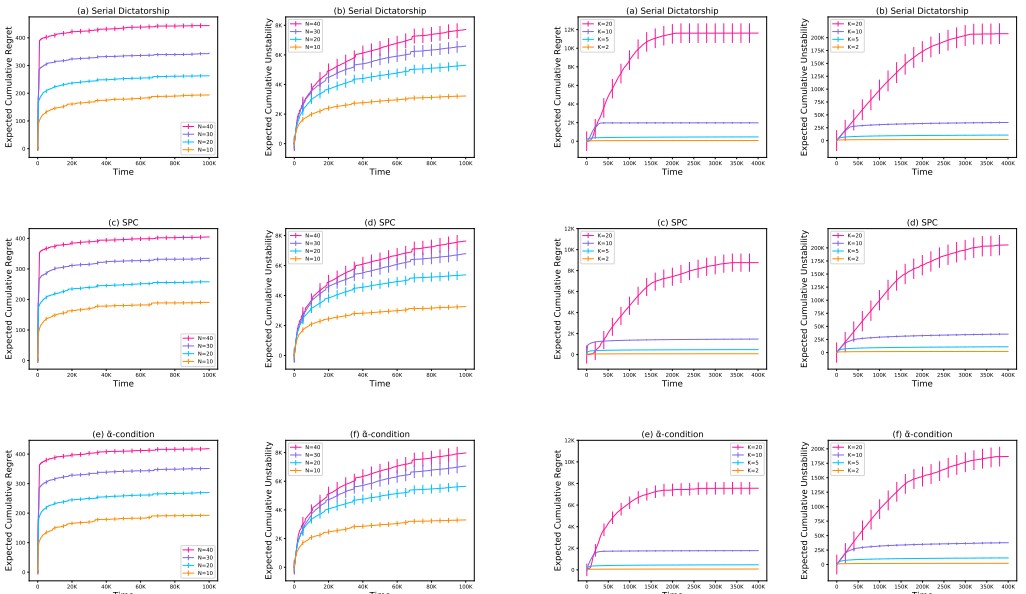

Figure 2: Cumulative regret and cumulative unstability of MO-UCB-D4 of size with $N \in \{10, 20, 30, 40\}$ and the number of arms $K = 5$ under *Serial Dictatorship*, *SPC*, $\tilde{\alpha}$-*condition*.

Figure 3: Cumulative regret and cumulative unstability of MO-UCB-D4 of size with $K \in \{2, 5, 10, 20\}$ under *Serial Dictatorship*, *SPC*, $\tilde{\alpha}$-*condition*.

326 **Varying arm capacity**   The number of arms $K$ is chosen by $K \in \{2, 5, 10, 20\}$, with $N = 20$ and
327 $q = N/K$. The number of rounds we set is $400,000$. With the increase of $K$, both the cumulative
328 regret in Figure 3(a)(c)(e) and the cumulative unstability in Figure 3(b)(d)(f) increase monotonously.
329 When $K$ increases, the capacity $q_k$ for each arm $k$ decreases, and then the number of collisions
330 will increase, which leads to an increase of cumulative regret. And it also leads to more unstable
331 pairs, which needs more communication blocks to converge to a stable matching. Under these three
332 conditions, the performances of the algorithm are similar.

## 7   Conclusion

334 We are the first to study the bandit algorithm for the many-to-one matching market under the unique
335 stable matching. This work focuses on a decentralized market. A new $\tilde{\alpha}$-*condition* is proposed
336 to guarantee a unique stable outcome in many-to-one market, which is more general than existing
337 uniqueness conditions like *SPC*, *Serial Dictatorship* and could recover the usual $\alpha$-*condition* in
338 one-to-one setting. We propose a phase-based algorithm of MO-UCB-D4 with arm-elimination,
339 which obtains $O\left(\frac{NK\log(T)}{\Delta^2}\right)$ stable regret under $\tilde{\alpha}$-*condition*. By carefully defining a mapping from
340 arms to the least preferred agent in its stable matched set, we could effectively correspond arms and
341 agents by individual-to-individual. A series of experiments under two environments of varying the
342 market size and varying arm capacity are conducted. The results show that our algorithm performs
343 well under *Serial Dictatorship*, *SPC* and $\tilde{\alpha}$-*condition* respectively.

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
