# OpenReview forum: "Bandit Learning in Many-to-one Matching Markets with Uniqueness Conditions"
_NeurIPS.cc/2022/Conference — NeurIPS 2022 Submitted_

### Official Review · Reviewer_HrNG · 2022-07-10

**Rating:** 5
**Confidence:** 4
**Soundness:** 3 good
**Presentation:** 3 good
**Contribution:** 2 fair

**Summary:**

The authors study the problem of minimizing regret in many-to-one matching markets with bandit feedback. In a many-to-one matching markets, each right-side agent (aka arms) can match upto a maximum number of left-side agents (aka agents). The authors develop \tilde{alpha}-condition as a sufficient condition for uniqueness consistency in many-to-one matching markets. Under this condition they provide logarithmic regret for the bandit learning problem for the MO-UCB-D4 algorithm.

**Questions:**

- The authors' comments on increase in q_k decreases lr_{max}(j) seems a bit less rigorous. Is there a more rigorous statement -- e.g. Does partial ordering of q_k introduces partial order on lr_{max}(j)?

- Please discuss the other points mentioned in the weaknesses.

**Limitations:**

This work seems theoretical in nature, and negative societal impact, if any, is not immediate.

**Strengths And Weaknesses:**

Strengths:
- This work is an interesting addition to learning in matching markets with bandit feedback, as it studies the many-to-one matching markets for the first time.
- It develops the \tilde{\alpha}-condition (adapted from one-to-one matching markets) as a sufficient and necessary condition for uniqueness consistency.

Weaknesses:
- The relationship of capacity q_k and  lr_{max}(j) is not discussed properly, elaboration needed (see questions).
- Counterexample of Acyclicity* holding and \tilde{\alpha}-condition not holding is missing. This will improve the understanding in section 4.3.
- The \tilde{\alpha} condition is also very closely related to Karpov et al. The difficulty in extending this to the many-to-one matching is unclear.
- The discussion on why deriving regret bound for many-to-one case is difficult than Basu et al. is not stated clearly. Specifically, the role of moving from ordering of individuals to ordering of sets need to be elaborated (probably in derivation of the \tilde{\alpha} condition as well).

---

> ### Author Response · Authors · 2022-08-02
> **Reply to Reviewer HrNG (part 1)**
>
> Thanks for your valuable comments. Please find our responses below.
>
> ### The definition of $lr_{\max}(j)$ and its relationship with the capacity $q_k$
>
> To give an intuitive representation of these two order, we define a mapping $lr$ to connect them. $lr(i) = \max \lbrace j:  A_j \in \gamma^{\ast}(m^{\ast}(i)) \rbrace $. And $lr_{\max}(j) = \max \lbrace lr(j'): 1\leq j' \leq j \rbrace $, is the maximum $lr$ mapping for agent $j'$ who ranks higher than $j$.
>
> The definition of $lr$ as above is affected by capacity. Since each arm can accommodate multiple agents, its matching result is a set. We want to give the range of the matched agents set of each arm under the stable matching, and $lr$ transforms the non one-to-one relationship into the correspondence between individuals, thus characterizing the threshold of the matched agents set. In addition, $lr$ is also related to the many-to-one market structure with capacity $q_k$, where the left order and the right order determines the sequence number of each stable matched agent, thus determining the $lr_{\max}$.
>
>
> ### Counterexample of $Acyclicity^{\ast}$ but not $\tilde{\alpha}$-condition
>
> Here we give an example that the preference satisfies $Acyclicity^{\ast}$ but not $\tilde{\alpha}$-condition in the following table.
>
> |  arm      |                                                     |
> | ---------- |  ------------------                             |
> | $c_1:$   | $s_1 > s_2 > s_5 > s_3 > s_4 $ |
> | $c_2:$   | $s_2 > s_1 > s_4 > s_3 > s_5 $ |
> | $c_3:$   | $s_1 > s_3 > s_2 > s_4 > s_5 $ |
>
>
> |  agent    |                                 |
> | ---------- |  ------------------          |
> | $s_1:$  | $ c_2 > c_3 > c_1 $ |
> | $s_2:$  | $ c_1 > c_2 > c_3 $  |
> | $s_3:$  | $ c_3 > c_1 > c_2 $  |
> | $s_4:$  | $ c_1 > c_2 > c_3 $  |
> | $s_5:$  | $ c_1 > c_2 > c_3 $  |
>
>
> From Table above, we now explain that a market with arms $c_1, c_2, c_3$, agents $s_1, s_2, s_3, s_4, s_5$, and capacity $q = (2, 1, 2)$ with arm preference and agents preference satisfies $Acyclicity^{\ast}$ and can lead to a unique stable matching but does not satisfy $\tilde{\alpha}$-condition. We run GS Algorithm in many-to-one market and obtain stable matching $\lbrace (c1; s_2, s_5), (c_2; s_1), (c_3; s_3, s_4) \rbrace$. And $Acyclicity^{\ast}$ is easily verified. After eliminating $(c_3; s_3, s_4)$, only $s_1, s_2, s_5, c_1, c_2$ remain in the system, and then the preference profile is represented as arm preference after deletion and agent preference after deletion in the table above.
>
>
> First we prove that it satisfies $Acyclicity^{\ast}$.
> From preference profile, we can find  at least ten cycle:
>
> 1. $s_1 \succ_{c_3} s_2 \succ_{c_2} s_1$;
> 2. $s_1 \succ_{c_1} s_2 \succ_{c_2} s_1$;
> 3. $s_3 \succ_{c_1} s_4 \succ_{c_2} s_3$;
> 4. $s_3 \succ_{c_3} s_4 \succ_{c_2} s_3$;
> 5. $s_2 \succ_{c_2} s_3 \succ_{c_3} s_2$;
> 6. $s_2 \succ_{c_1} s_3 \succ_{c_3} s_2$;
> 7. $\cdots$
>
>
> Condition $(P)$ in Definition 11 is satisfied, and we then illustrate that condition $(Q)$ is not satisfied, thus $Acyclicity^{\ast}$ holds. For cycle (i), $T_1, T_2 \subset S \backslash \lbrace s_1, s_2 \rbrace $, $|T_1| = q_{c_1} - 1 = 1$. However, it violates $T_1 \subset U_{c_1}(s_1) = \emptyset$.
>
> Next we prove that it does not satisfy $\tilde{\alpha}$-condition.
>
>
> | arm  |                                  |
> | ------  |  -------------------------     |
> | $c_1:$  | $s_1 > s_2 > s_5 $ |
> | $c_2:$  | $s_2 > s_1 > s_5 $ |
>
>
> |  agent |                           |
> | ------- |  ---------------------   |
> |   $s_1:$ | $ c_2 > c_1 $    |
> |   $s_2:$ | $ c_1 > c_2 $    |
> |   $s_5:$ | $ c_1 > c_2 $    |
>
>
> After eliminating $(c_3; s_3, s_4)$, only $s_1, s_2, c_1, c_2$ remain in the system, and then the preference profile is represented as arm preference after deletion and agent preference after deletion in the table above. Apparently, this preference can produce two stable matching. Thus, $\tilde{\alpha}$-condition is violated.
>
> ### Difficulties in the extension of many-to-one setting
>
> First, in our decentralized setting, collision among agents is one of main causes of regret, while capacity will hinder the collision-reducing process. In communication block, when two agents select one arm at the same time, as an arm can accept more than one agent at a time, these two cannot distinguish who is more preferred by this arm, which can be done in one-to-one markets as Basu et al. Thus it is more difficult to identify arms' preferences for each agent.
> Second, considering many-to-one case, the matching result for each arm $k$ is a set.
> The $lr$ in Basu et al. is a one-to-one mapping that corresponds the agent index in the left order and the agent index in the right order, which is related to regret bound (Theorem 3 in Basu et al. and Theorem 2 in our work).
> While it does not hold in our setting.
> To give a descriptive range of matched result for each arm under $\tilde{\alpha}$-condition, we need to define a new mapping.

---

> > ### Author Response · Authors · 2022-08-02
> > **Reply to reviewer HrNG (part 2)**
> >
> > ### Difficulties in extending $\alpha$-condition to the $\tilde{\alpha}$-condition in many-to-one matching
> >
> > To extend $\alpha$-condition to the many-to-one setting, it needs to define the preferences among sets. However, there might be exponential number of sets due to the combinatorial structure and simply constraining preferences over all possible sets will lead to high complexity. Also, with the constraints on preferences over all possible sets, it would be complicated and difficult to limit the range of blocking agents according to the key left and right orders in the regret analysis. Thus based on the complexity and analysis requirement, it would be better to scale down the constraint range of the preferences. Motivated by the definition of $\alpha$-condition which characterizes properties of matched pairs in one-to-one setting, we come up with a possible constraint by regarding the arm and its least preferred agent in the matched set as the *matched pair* and define preferences according to this grouping. It turns out that we only need to define the preferences of arms over disjoint sets of agents to complete the extension, which can also fit the regret analysis well. As a summary, there might be other possible ways to extend the $\alpha$-condition but we present a successful trial to not only give a good extension with similar inclusion relationships but also guarantee good regret bound.
> >
> > ### References
> >
> > [1] Orly Avner and Shie Mannor. Concurrent bandits and cognitive radio networks. In Joint European Conference on Machine Learning and Knowledge Discovery in Databases, pages 66–81. Springer,2014.
> >
> > [2] Soumya Basu, Karthik Abinav Sankararaman, and Abishek Sankararaman. Beyond $\log^2(t)$ regret for decentralized bandits in matching markets. In International Conference on Machine Learning, pages 705–715, 2021.
> >
> > [3] Simon Clark. The uniqueness of stable matchings. Contributions in Theoretical Economics, 6(1), 2006.
> >
> > [4] Lydia T Liu, Horia Mania, and Michael Jordan. Competing bandits in matching markets. In International Conference on Artificial Intelligence and Statistics, pages 1618–1628. PMLR,2020.
> >
> > [5] Lydia T Liu, Feng Ruan, Horia Mania, and Michael I Jordan. Bandit learning in decentralized matching markets. arXiv preprint arXiv:2012.07348, 2020.
> >
> > [6] Jonathan Rosenski, Ohad Shamir, and Liran Szlak. Multi-player bandits–a musical chairs approach. In International Conference on Machine Learning, pages 155–163. PMLR, 2016.
> >
> > [7] Abishek Sankararaman, Soumya Basu, and Karthik Abinav Sankararaman. Dominate or delete: Decentralized competing bandits in serial dictatorship. In International Conference on Artificial Intelligence and Statistics, pages 1252–1260. PMLR, 2021.

---

> > > ### Comment · Reviewer_HrNG · 2022-08-07
> > > **Response to Rebuttal**
> > >
> > > I thank the authors for their response to the reviews. The responses are satisfactory, and the authors are requested to add these in the paper.

---

> > > > ### Author Response · Authors · 2022-08-07
> > > > **Reply to reviewer HrNG**
> > > >
> > > > Thanks for your comments, and we will add these contents to our paper.

---

### Official Review · Reviewer_Q3cx · 2022-07-12

**Rating:** 3
**Confidence:** 3
**Soundness:** 2 fair
**Presentation:** 1 poor
**Contribution:** 2 fair

**Summary:**

This paper studies decentralized learning and matching in many-to-one matching markets. They introduce a model for this setting, develop a bandit learning algorithm under a uniqueness assumption about the set of stable matchings, and obtain a logarithmic instance-dependent bound on its regret.

**Questions:**

- I was not able to understand the definitions of “left order” and “right order”; it seems like there is a missing clause in the first sentence of the paragraph starting at line 173. Would it be possible to elaborate further on the definition?
- In Definition 2, there appears to be a missing term in the summations. What should that term be?
- I am also somewhat confused about the comparisons between assumptions on the preferences under “Conditions for Unique Stable matching”. For instance, the condition of Akahoshi is a property of the preference profiles of a single side (under which there exists a unique stable matching). On the other hand, uniqueness consistency is a property of the full preference profiles (for both sides). Could you clarify what the precise comparison being made is?

**Limitations:**

The authors describe the limitations of their results in Section C.3. It may be useful to move this discussion into the main body.

**Strengths And Weaknesses:**

- Strengths:
    - The problem of many-to-one matching that the authors study is well-motivated, as it is a salient aspect of many real-world matching markets (e.g., schools, employment) in which learning and adaptivity could play a useful role.
    - It is nice to see simulations of the algorithm to validate the performance. (However, it would also be nice to see comparisons against other methods.)
- Weaknesses:
    - The writing was difficult to follow, with several crucial definitions and assumptions unclear (see below).
    - The focus on unique stable matching seems like a strong limitation of the results, as in many markets where the preferences are not in a sense “acyclic” or “aligned”, multiple stable matchings exist. The proposed approach does not appear to scale gracefully to this setting.
    - While the consideration of many-to-one matching is new, this work only considers a limited form of many-to-one matching where each “arm” has a total preference ordering agents.
    - The regret bounds are in the minimum reward gap across all agents in arms; this assumption does not permit agents to be indifferent (or nearly indifferent), even for arms far down in their ranking list.

---

> ### Author Response · Authors · 2022-08-02
> **Reply to reviewer Q3cx (part 1)**
>
> Thanks for your valuable comments. Please find our responses below.
>
> ### $\Delta$ in the regret bound
>
> First we would like to clarify the exact definition of $\Delta$. The reward gap for agent $j$ and arm $k$ is defined as $\Delta_{jk} = |\mu_{j, m^{\ast}(j)} - \mu_{j,k}|$, which is the reward gap between arm $k$ and the stable matched arm $m^*(j)$ for agent $j$. And $\Delta$ is the minimum reward gap of all arms for all agent $j$, which is defined as $\Delta = \min_{j \in [N]}\{ \min_{k \in [K]} \Delta_{j,k} \}$. Note that we have assumed $\mu_{j,k} \neq \mu_{j,k'}$ for any $k \neq k'$ and $j \in [N]$.
>
> Note that this setting involves multi-players and one agent's preference change could result in a totally different stable matching. Thus it is necessary to have $\Delta$ appear in the regret bound. Also the lower bound of $O(\frac{\log(T)}{\Delta^2})$ in the one-to-one setting with uniform arm preferences (a special case of ours) [7] also shows the term $\Delta$ is unavoidable. Note that this does not mean agents are indifferent. Actually we allow agents to have different preferences but the minimal gap serves as the bottleneck for the regret hardness.
>
> ### Definitions of *the left order*, *the right order*
>
> The left order and the right order are both permutations of agent index and arm index. Denote $[N], [K]$ as the left order and $[N]_r,[K]_r$ as the right order. The $k$-th arm in the right order set $[K]_r$ has the index $c_k$ in the left order set $[K]$ and the $j$-th agent in the right order set $[N]_r$ has the index $A_j$ in the left order set $[N]$. From our definition of $\tilde{\alpha}$-condition, the left order satisfies
>
> $\forall j\in[N], \forall k>j, k\in[K], \mu_{j, m^{\ast}(j)} > \mu_{j, k} $
>
> where $m^{\ast}(j)$ is agent $j$'s stable matched arm, and the right order satisfies
>
> $ \forall k< k' \leq K,$ $ c_k \in [K]_r,$
>
> $ A_{k' }\subset [N]_r,  $
>
> $\gamma^{\ast}(c_k) \succ_{c_k} A_{\sum_{i=1}^{k'-1} q_{c_i} +1} $
>
> where the agent set $\gamma^{\ast}(c_k)$ is more preferred than agent $A_{\sum_{i=1}^{k'-1} q_{c_i} +1}$ means that the least preferred agent in $\gamma^{\ast}(c_k)$ for arm $c_k$ is better than agent $A_{\sum_{i=1}^{k'-1} q_{c_i} +1}$ under the preference of arm $c_k$.
>
> The left order gives rankings according to agents' preferences. The first agent in the left order set $[N]$ prefers arm $1$ in $[K]$ most and has it as the stable matched arm. Similar properties for the agent $2$ to $q_1$ since arm $1$ has $q_1$ capacity. Then the $(q_1 +1)$-th agent in the left order set $[N]$ has arm $2$ in $[K]$ as her stable matched arm and prefers arm $2$ most except arm $1$. The remaining agents follow similarly.
>
> The right order gives rankings according to arms' preferences.  The first arm $1$ in the right order set $[K]_{r}$ most prefers first
>
> $q_{c_1}$ agents in the right order set $[N]_r$ and take them as its stable matched agents. The remaining arms follow similarly.
>
> The concepts of the left order and right order might be a bit abstract, thus we have presented an illustrative figure (Figure 4) in Appendix A.1. Perhaps this is a bit far away from their appearances in the main body. We would rearrange our paper to make it easier to understand.
>
> ### The assumption of the unique stable matching
>
> When the preferences of agents and arms are given by some utility functions instead of random preferences, like the payments for workers in the labor markets, the stable matching is usually unique. Thus the assumption of the unique stable matching is quite common in real applications. Besides, some uniqueness conditions have important properties like consistency, which states that any stable pair leaving the market does not affect the remaining stable matching. In dynamic markets where agents and arms come and go, the consistency property is desirable to keep the matching majority static [2]. Furthermore, when the stable matching is unique, there would be no dispute about adopting stable matching preferred by which side, thus is more fair to both sides. Note that the outcome of the GS algorithm would prefer the proposal side and would be unfair to the other side [3].

---

> > ### Author Response · Authors · 2022-08-02
> > **Reply to reviewer Q3cx (part 2)**
> >
> > ### The Akahoshi condition
> >
> > First, the condition of Akahoshi is actually the $Acyclicity^{\ast}$ condition discussed in our appendix. It is indeed a more general condition than our $\tilde{\alpha}$-condition to guarantee a unique stable matching. Although our $\tilde{\alpha}$-condition is relatively strong, it has an important property of consistency, which guarantees unique stable matching after possible deletions of stable pairs and is a key property to design our algorithm. It guarantees that our algorithm could find a good global unique stable matching in such an iterative manner with good regret guarantee. Without the property of consistency (like Akahoshi condition), it is not clear how to design an efficient algorithm. We would leave this as an interesting future work.
> >
> > ### Simulations compared with other methods
> >
> > While the main focus of the paper is theory and therefore we did not put much emphasis on the experimental evaluation, we still carefully design our experiments to test our algorithm across different environments. Since our work is the first one to study the many-to-one setting, there is indeed no comparable baselines. We mainly follow the experimental settings of previous works [4, 5] which also do not have suitable baselines and mainly test the performances of their own algorithms on different environments. Furthermore, please note that in the experiments our goal was to demonstrate the robustness of our algorithm and we think that the experimental results do achieve this goal.
> >
> > ### The missing term in Definition 2
> >
> > Thanks for pointing this out. There is a missing term of the capacity $q_{c_i}$. And we give the complete definition in the following.
> >
> > **Definition 2**
> >
> > A many-to-one matching market satisfies the *$\tilde{\alpha}$-condition* if,
> >
> > (i) The left order of agents and arms satisfies
> > $$\forall j\in[N], \forall k>j, k\in[K], \mu_{j, m^{\ast}(j)} > \mu_{j, k}, $$
> > where $m^{\ast}(j)$ is agent $j$'s stable matched arm;
> >
> >
> > (ii) The right order of agents and arms satisfies
> >
> > $  \forall k< k' \leq K$,
> >
> > $c_k \in [K]_{r}$,
> >
> > $A_{k'} \subset [N]_{r}$,
> >
> > $\gamma^{\ast}(c_k) \succ_{c_k} A_{\sum_{i=1}^{k'-1} q_{c_i} +1}, $
> >
> > where the set $\gamma^{\ast}(c_k)$ is more preferred than $A_{\sum_{i=1}^{k'-1} q_{c_i} +1}$ means that the least preferred agent in $\gamma^{\ast}(c_k)$ for $c_k$ is better than $A_{\sum_{i=1}^{k'-1} q_{c_i} +1}$ for $c_k$.
> >
> >
> > ### Organization of our paper
> > Thanks for your suggestions and we will rearrange our paper accordingly.
> >
> > ### References
> >
> > [1] Orly Avner and Shie Mannor. Concurrent bandits and cognitive radio networks. In Joint European Conference on Machine Learning and Knowledge Discovery in Databases, pages 66–81. Springer,2014.
> >
> > [2] Soumya Basu, Karthik Abinav Sankararaman, and Abishek Sankararaman. Beyond $\log^2(t)$ regret for decentralized bandits in matching markets. In International Conference on Machine Learning, pages 705–715, 2021.
> >
> > [3] Simon Clark. The uniqueness of stable matchings. Contributions in Theoretical Economics, 6(1), 2006.
> >
> > [4] Lydia T Liu, Horia Mania, and Michael Jordan. Competing bandits in matching markets. In International Conference on Artificial Intelligence and Statistics, pages 1618–1628. PMLR,2020.
> >
> > [5] Lydia T Liu, Feng Ruan, Horia Mania, and Michael I Jordan. Bandit learning in decentralized matching markets. arXiv preprint arXiv:2012.07348, 2020.
> >
> > [6] Jonathan Rosenski, Ohad Shamir, and Liran Szlak. Multi-player bandits–a musical chairs approach. In International Conference on Machine Learning, pages 155–163. PMLR, 2016.
> >
> > [7] Abishek Sankararaman, Soumya Basu, and Karthik Abinav Sankararaman. Dominate or delete: Decentralized competing bandits in serial dictatorship. In International Conference on Artificial Intelligence and Statistics, pages 1252–1260. PMLR, 2021.

---

> > > ### Author Response · Authors · 2022-08-07
> > > **Reply to reviewer Q3cx**
> > >
> > > Thanks again for your valuable comments. Since your initial attitude is negative, please let us know if you have any concerns about our reply or any further questions. As the reviewer-author discussion period is coming to an end, we sincerely hope to have more communication before the end of the discussion.

---

> > > > ### Comment · Reviewer_Q3cx · 2022-08-09
> > > > **.**
> > > >
> > > > Thanks for the response! I appreciate the clarifications; however, I do feel that the presentation has significant room for improvement (e.g. incorporating the above clarifications and corrections). Regarding the $\Delta$, while I agree that previous works have shown its necessity in the rate, it seems to point to a limitation of this definition of regret: assuming away indifferences (w.r.t. one's stable match) does not feel natural from a matching markets perspective.

---

> > > > > ### Author Response · Authors · 2022-08-09
> > > > > **Clarifications on the ``indifferent agents''**
> > > > >
> > > > > We apologize we have misunderstood your question on the ``minimum reward gap''. We agree that the case of indifferent agents would be more general. However, as far as we know, a lot of works studying the traditional (offline) matching markets would assume preferences to be strict [3, 8, 9, 10, 11], perhaps due to the reason of simplicity. Our work mainly follows these existing settings of the (offline) matching markets [3, 8, 9, 10, 11] and the bandit learning on the one-to-one matching markets [2, 4, 5, 7] that assume strict preferences.
> > > > >
> > > > > Note that if the agents are indifferent (or nearly indifferent) over the arms that are far down the ranking lists and do not affect the stable matching, our algorithm and analysis can actually go through. The gap that appeared in the regret bound actually depends only on the those ``(nearly) optimal'' arms that appear in the stable matching or are the best among those not appeared in the stable matching.
> > > > >
> > > > >
> > > > > Recall that our setting is to learn a particular stable matching, like previous works [2, 4, 5, 7] learning the unique, or agent-pessimal/optimal stable matching on the one-to-one setting. Under this objective, if the agents are nearly indifferent, not exactly indifferent, over ''(nearly) optimal'' arms, no matter how small the gap is, the agents will need to figure out which arm is better and the gap appears as the learning hardness. This phenomenon is common in multi-armed bandits where differentiating the optimal arm and the second optimal arm is the most difficult part of the learning. Then one might be curious about the objective to learn a ``nearly stable matching''. We agree this would be more general and would prefer to leave it as interesting future work.
> > > > >
> > > > > For the case when agents are exactly indifferent on ''(nearly) optimal'' arms, the stable matchings would not be unique. In this case, the communication block and the global deletion set of our algorithm need to be revised to allow each agent to keep more than one stable matched arm. Note that after this revision, the selected matching will not become fixed during interactions and will switch between all optimal stable matchings since the learning algorithm needs to continue exploring these arms to take precautions against the case of a small gap. This will result in a phenomenon of fast-changing matching-selections, compared with our setting and most previous works [2, 4, 5, 7] where the learning algorithm tends to stick to a specific matching in the latter learning period.
> > > > >
> > > > > Thanks for your valuable comments. We have finished the main revisions suggested by you and other reviewers, where the updated pdf has been uploaded to the system, and we promise to continue polishing our paper to make it clearer. We hope the above reply could address your concern and are willing to answer any further questions before the discussion ends.
> > > > >
> > > > >
> > > > >
> > > > > [1] Orly Avner and Shie Mannor. Concurrent bandits and cognitive radio networks. In Joint European Conference on Machine Learning and Knowledge Discovery in Databases, pages 66–81. Springer,2014.
> > > > >
> > > > > [2] Soumya Basu, Karthik Abinav Sankararaman, and Abishek Sankararaman. Beyond $\log^2 (t)$ regret for decentralized bandits in matching markets. In International Conference on Machine Learning, pages 705–715, 2021.
> > > > >
> > > > > [3] Simon Clark. The uniqueness of stable matchings. Contributions in Theoretical Economics, 6(1), 2006.
> > > > >
> > > > > [4] Lydia T Liu, Horia Mania, and Michael Jordan. Competing bandits in matching markets. In International Conference on Artificial Intelligence and Statistics, pages 1618–1628. PMLR,2020.
> > > > >
> > > > > [5] Lydia T Liu, Feng Ruan, Horia Mania, and Michael I Jordan. Bandit learning in decentralized matching markets. arXiv preprint arXiv:2012.07348, 2020.
> > > > >
> > > > > [6] Jonathan Rosenski, Ohad Shamir, and Liran Szlak. Multi-player bandits–a musical chairs approach. In International Conference on Machine Learning, pages 155–163. PMLR, 2016.
> > > > >
> > > > > [7] Abishek Sankararaman, Soumya Basu, and Karthik Abinav Sankararaman. Dominate or delete: Decentralized competing bandits in serial dictatorship. In International Conference on Artificial Intelligence and Statistics, pages 1252–1260. PMLR, 2021.
> > > > >
> > > > > [8] David Gale and Lloyd S Shapley. College admissions and the stability of marriage. The American Mathematical Monthly, 69(1):9–15, 1962.
> > > > >
> > > > > [9] Alexander Karpov. A necessary and sufficient condition for uniqueness consistency in the stable marriage matching problem. Economics Letters, 178:63–65, 2019
> > > > >
> > > > > [10] Takashi Akahoshi. Singleton core in many-to-one matching problems. Mathematical Social Sciences, 72:7–13, 2014
> > > > >
> > > > > [11] Hai Nguyen, Thành Nguyen, and Alexander Teytelboym. 2021. Stability in matching markets with complex constraints. Management Science 67, 12 (2021), 7438–745

---

### Official Review · Reviewer_36px · 2022-07-18

**Rating:** 6
**Confidence:** 2
**Soundness:** 3 good
**Presentation:** 2 fair
**Contribution:** 3 good

**Summary:**

This paper studies the problem of bandit learning in many-to-one matching markets. They generalize the $\alpha$-condition in a marriage problem to a proposed $\tilde\alpha$-condition that is a matching guarantee for uniqueness consistency in many-to-one matching markets. Under the proposed condition the authors present a UCB-based arm elimination algorithm that obtains logarithmic regret . The authors verify their proposed algorithm by simulations under several different optimality conditions.

**Questions:**

Can the authors address some of the queries from the previous section? Specifically, can they contrast their work from prior work in terms of (i) theory and regret bound, (ii) novel components in the regret analysis, (iii) specific shortcomings in prior work, (iv) baselines for experiments, (v) optimality of regret bound?

**Limitations:**

The authors do not discuss limitations.

**Strengths And Weaknesses:**

Strengths:
+ The paper presents a novel algorithm with a competitive regret bound.
+ The new $\tilde\alpha$-condition proposed appears to generalize the $\alpha$-condition from the stable marriage problem, and the discussion for the condition puts it in comparison with prior work.

Weaknesses:
- The paper has sections that are not very well-written. While the authors do a good job of explaining their new stability condition in Section 4.1, the discussion with prior conditions (deferred to 4.3) should essentially be put together. In Section 5, the authors present the technical challenges in extending bandit matching to the many-to-one setting, however it is unclear / difficult to follow how each of the steps translate into the final regret bound which is presented earlier in Section 4.2.
- Regret Bound: It is not clear which contributions are novel and which proof techniques are reused from prior work. All the proofs are deferred to the appendix, and the proof-sketch provided in Section 4.2 is not very informative.
- - There are no discussions on which parts of the proof reuse prior work and which parts are novel.
- - There are no discussions on optimality and lower bounds beyond the mentioning of the logarithmic factor.
- - There are no discussions on which aspects of the algorithm can be improved in future work, and what the authors conjecture to be the optimal rate.
- Experiments: While the authors indeed experiment with several different criteria and different number of agents, they do not have any baseline comparisons, making it difficult to compare their experimental results in a relative sense. It would be worthwhile if the authors could come up with some meaningful albeit sub-optimal baselines to highlight the importance of each component in their algorithm design.

---

> ### Author Response · Authors · 2022-08-02
> **Reply to reviewer 36px (part 1)**
>
> Thanks for your valuable comments. Please find our responses below.
>
> ### Optimality of our bound and the lower bound
>
> Recall that our bound is $O(NK\frac{\log(T)}{\Delta^2})$. There exists a lower bound of $O(\frac{\log(T)}{\Delta^2})$ under the setting where arms have the same and known preferences [7], which is a special case of our setting. Our bound is optimal in terms of $T$ and $\Delta$.
>
> For $N$, since each agent $j$ needs to face collisions from non-dominated arms and other agents, regret is bounded over the summation of agents and thus leads to the term $O(N)$.
> Usually in a multi-player decentralized setting [1, 6], each agent will suffer regret of term $N$ since it will be collided with other agents. Thus we conjecture such $N$ is unavoidable.
>
> For $K$, since in decentralized setting, agents have no knowledge of arm preference, each agent needs to try each $O(\log (T)/\Delta^2)$ times to identify the stable matched arm. And it may get collided when pulling the other agent's stable matched arm, thus leading to the term $K$. $K$ might be removed for those agents who may never get collisions due to special market structure.
>
>
> ### Novel components in the regret analysis
>
> First of all, we would like to emphasize that the $\tilde{\alpha}$-condition is new for unique stable matching in the many-to-one matching market, even in the traditional (offline) setting of matching markets without interactions. Moreover, to formulate this condition and verify its relationship with existing conditions are not trivial.
>
> Besides this new condition, we use a mapping from a set to a representative individual in order to tackle the transition from one-to-one to many-to-one setting.
> In our setting, each arm $k$ can accommodate more than one agent, the matched result for each arm in the right order is a set. To match agent index in the left order to agent index in the right order, we define a mapping $lr$ as the least preferred agent among matched agents set of this arm, which is formally represented as $lr(i) = \max \{j:  A_j \in \gamma^{\ast}(m^{\ast}(i)), j\in [N]\}$ (recall that $[N]_r,[K]_r$ are permutations of $[N]$ and $[K]$. The $k$-th arm in $[K]_r$ is the $c_k$-th arm in $[K]$, and the $j$-th agent in $[N]_r$ is the $A_j$-th agent in $[N]$.). It gives a descriptive range of matched result for each arm. While the mapping $lr$ under $\alpha$-condition [2] is a one-to-one relationship, which does not hold in our setting.
>
> The key point of blocking agents also needs to be carefully identified. Unlike one-to-one setting where agents can be easily compared, many-to-one setting might not provide direct comparisons since arms could accept many agents. After global deletion, the agents that still generate many collisions with agent $j$ are regarded as non-dominated blocking agents.
> Thus regret of local deletion block (Appendix A.3) is formed with this agent set.
>
> For the parts of the proof sketch of regret upper bound and the iterative idea, they are standard, and mainly follow existing works [2, 7]. We decompose regret into two blocks, before and after a *`good phase'*. After it, regret is caused by collision, sub-optimal arm, regret minimization, communication, and phases before it is bounded by an iterative manner.
>
>
> ### How each of the steps translate into the final regret bound
>
> The main proof idea is that we use induction to show how agents settle down to their stable matched arms. Agent $1$ will find its stable matched arm $1$ at first since arm $1$ is the most preferred arm for agent $1$. Then agent $2$ will find its stable arm $2$ since agent $2$ has deleted arm $1$ in the communication block and thus arm $2$ becomes its most preferred arm. We can show by induction that agent $j$ will find its stable matched arm after agent $1$ to $j-1$ has settled down.
>
> The regret of agent $j$ can be decomposed into sub-optimal play, collisions, communication and local deletion four parts. Both collisions between agent $j$ and other agents in the blocking agent set and sub-optimal play are due to the wrong estimation of UCB index (Lemma 6). Communication regret can be bounded by the length of communication block. Local deletion regret can be controlled by the threshold we set (line 6 in Algorithm 1).
>
> ### Limitations of prior work
> Previous works mainly study one-to-one matching with bandits, ignoring the combinatorial phenomenon which often occur in the real applications, like short-term recruitment on online platforms and school enrollment. Similarly, the existing techniques are mainly designed for one-to-one setting and can not be directly generalized. New setting of many-to-one needs to be studied and new methods that transform matched agent sets to an individual for arms needs to be developed.

---

> > ### Author Response · Authors · 2022-08-02
> > **Reply to reviewer 36px (part 2)**
> >
> > ### Aspects of the algorithm can be improved in future work
> >
> > Our algorithm adopts global deletion and local deletion to communication (line 15) and regret minimization block (line 16). These are based on two sets, globally dominated arm set and non-dominated blocking set. Currently the two blocks are dealt separately. Since constructing these two sets needs same information of identifying other agents' preferences, perhaps there is a more efficient deletion block that can deal with these two at the same time, thus to save blocking agent set efficiently and to reach a better regret.
> > In addition, index estimation algorithm for agents can be studied further.
> >
> > We will add more discussions in the next version.
> >
> > ### Baselines for experiments
> >
> > While the main focus of the paper is theory and therefore we did not put much emphasis on the experimental evaluation, we still carefully design our experiments to test the robustness of our algorithm across different environments. Since our work is the first one to study the many-to-one setting, there is indeed no comparable baselines. Furthermore, please note that in the experiments our goal was to demonstrate the strengths of our algorithm and we think that the experimental results do achieve this goal, as also done in one-to-one setting [4, 5].
> >
> > ### Organization of our paper
> >
> >  Thanks for your suggestions and we will rearrange our paper accordingly.
> >
> > ### References
> >
> > [1] Orly Avner and Shie Mannor. Concurrent bandits and cognitive radio networks. In Joint European Conference on Machine Learning and Knowledge Discovery in Databases, pages 66–81. Springer,2014.
> >
> > [2] Soumya Basu, Karthik Abinav Sankararaman, and Abishek Sankararaman. Beyond $\log^2(t)$ regret for decentralized bandits in matching markets. In International Conference on Machine Learning, pages 705–715, 2021.
> >
> > [3] Simon Clark. The uniqueness of stable matchings. Contributions in Theoretical Economics, 6(1), 2006.
> >
> > [4] Lydia T Liu, Horia Mania, and Michael Jordan. Competing bandits in matching markets. In International Conference on Artificial Intelligence and Statistics, pages 1618–1628. PMLR,2020.
> >
> > [5] Lydia T Liu, Feng Ruan, Horia Mania, and Michael I Jordan. Bandit learning in decentralized matching markets. arXiv preprint arXiv:2012.07348, 2020.
> >
> > [6] Jonathan Rosenski, Ohad Shamir, and Liran Szlak. Multi-player bandits–a musical chairs approach. In International Conference on Machine Learning, pages 155–163. PMLR, 2016.
> >
> > [7] Abishek Sankararaman, Soumya Basu, and Karthik Abinav Sankararaman. Dominate or delete: Decentralized competing bandits in serial dictatorship. In International Conference on Artificial Intelligence and Statistics, pages 1252–1260. PMLR, 2021.

---

> > > ### Comment · Reviewer_36px · 2022-08-08
> > > **Thank you to the authors, score increased to 6.**
> > >
> > > Thanks to the authors for addressing my concerns, the central contributions are more clear to me now. I have adjusted my score to account for this, and I expect the reviewers will make their contributions more concise in the updated version.

---

> > > > ### Author Response · Authors · 2022-08-08
> > > > **Reply to reviewer 36px**
> > > >
> > > > Thanks for increasing the score, and we will make our contributions clearer in the revision.

---

### Official Review · Reviewer_zkK9 · 2022-07-29

**Rating:** 7
**Confidence:** 3
**Soundness:** 3 good
**Presentation:** 3 good
**Contribution:** 3 good

**Summary:**

The paper studies many-to-one matching markets - which is a generalization of the one-to-one matching markets studied in recent years in the online learning community. The paper identifies a combinatorial condition that guarantees an unique Nash equilibrium to the underlying game if all participants knew of their preferences. This condition is a multi-way generalization of the recently used alpha condition for one-to-one matching. The paper shows that if the underlying market satisfies this unique Nash Equilibria criteria, then the decentralized learning algorithm yields logarithmic regret. The proposed algorithm is a natural variant of the UCB-D4 which was shown to be a good method for the one-to-one matching case.

**Questions:**

1. What do you think are the additional challenges in extending this to markets with multiple equilibria ?
2. Can this algorithm be generalized if both sides of the market need to perform learning ?

**Limitations:**

Yes. This is primarily a theory paper.

**Strengths And Weaknesses:**

Strengths

1. Comprehensive treatment of the problem
2. Builds and contributes to an emerging theory on decentralized bandit learning in matching markets
3. Their algorithms and results recover the known results in one-to-one matching market when the system is one-to-one. In this sense the proposed algorithm is a strict generalization of prior algorithms

Weakness

1. The paper does not address issues such as competition mechanism while learning.

However, I believe the weakness to be minor and can be easily addressed in the discussion sections of the revision of the paper.

---

> ### Author Response · Authors · 2022-08-02
> **Reply to reviewer zkK9**
>
> Thanks for your valuable comments. Please find our responses below.
>
> ### Additional challenges in extending this to markets with multiple equilibria
>
> In this paper, we consider the $\tilde{\alpha}$-condition to guarantee the uniqueness consistency that any subset of market individuals leaving the system has no effect on the unique stable matching of the whole market. This property makes it possible to set the arm-deletion process to delete those 'arms preferred by the higher ranked agents' for each agent $j$. While in the multiple equilibria, we don't have this uniqueness consistency property and thus the algorithm may not find the stable matching like previous work [2]. There exist some works to deal with multiple equilibria in one-to-one setting [4, 5] and they suffer more regret than our algorithm. It might be possible to extend those algorithms to the many-to-one matching with multiple equilibria.
>
>
> ### Can this algorithm be generalized if both sides of the market need to perform learning?
>
> It might be possible that our algorithm can be extended to that both sides perform learning, but there exist some difficulties to handle. The core idea is that when one side learns, the other side will stop learning and give feedback to the learning side. We can set the algorithm with multi-phases, and each phase contains two main blocks. In the first block, each agent runs our algorithm to estimate the preference over agents, while each arm $k$ will select its top $q_k$ agents from its estimated preference in the last phase. In the second block, the arm side learns and agents take actions based on their estimated preference in the last phase. However, it is hard to control the regret during the learning process in this algorithm since the wrong estimation of one side in the last phase will give wrong feedback to the other side. The error might be accumulated and hard to control.
>
> ### References
>
> [1] Orly Avner and Shie Mannor. Concurrent bandits and cognitive radio networks. In Joint European Conference on Machine Learning and Knowledge Discovery in Databases, pages 66–81. Springer,2014.
>
> [2] Soumya Basu, Karthik Abinav Sankararaman, and Abishek Sankararaman. Beyond $\log^2(t)$ regret for decentralized bandits in matching markets. In International Conference on Machine Learning, pages 705–715, 2021.
>
> [3] Simon Clark. The uniqueness of stable matchings. Contributions in Theoretical Economics, 6(1), 2006.
>
> [4] Lydia T Liu, Horia Mania, and Michael Jordan. Competing bandits in matching markets. In International Conference on Artificial Intelligence and Statistics, pages 1618–1628. PMLR,2020.
>
> [5] Lydia T Liu, Feng Ruan, Horia Mania, and Michael I Jordan. Bandit learning in decentralized matching markets. arXiv preprint arXiv:2012.07348, 2020.
>
> [6] Jonathan Rosenski, Ohad Shamir, and Liran Szlak. Multi-player bandits–a musical chairs approach. In International Conference on Machine Learning, pages 155–163. PMLR, 2016.
>
> [7] Abishek Sankararaman, Soumya Basu, and Karthik Abinav Sankararaman. Dominate or delete: Decentralized competing bandits in serial dictatorship. In International Conference on Artificial Intelligence and Statistics, pages 1252–1260. PMLR, 2021.

---

### Meta-Review · Area_Chair_eJV6 · 2022-08-25

**Recommendation:** Reject
**Confidence:** Certain

**Metareview:**

The paper generalizes one of the results of Basu et al.'s ICML'21 paper to the case of many-to-one matchings, showing a distributed multi-armed bandits mechanism with logarithmic regret for finding a stable matching assuming a strong uniqueness condition on the input. I believe the paper does not meet the bar for acceptance into NeurIPS due to the following major weaknesses:
* The paper is not very well written and it is hard to understand in places.
* It is not clear how difficult the generalization is, given the existing result in the one-to-one matching case.
* The assumptions of the model, in particular the uniqueness assumption, are very strong, and restrict the result to a very limited (and fragile) set of input instances.

**Award:**

No

---

### Decision · Program_Chairs · 2022-09-14

Reject